# LLM-Datasets: An Open Framework for Pretraining Datasets of Large Language Models

**Malte Ostendorff**[1,2]**, Pedro Ortiz Suarez**[3]**, Lucas Fonseca Lage**[4]**, Georg Rehm**[4,1]
[1]Occiglot, [2]Deutsche Telekom, [3]Common Crawl Foundation,
[4]Deutsches Forschungszentrum für Künstliche Intelligenz GmbH (DFKI)
{firstname}@occiglot.org

## Abstract

Large language models have become the cornerstone of today's natural language processing research. To facilitate the training, evaluation, and deployment of language models, the community has developed a series of tools and frameworks and made them openly available. This joint community effort has led to more collaboration, standardization, and overall more progress in language model research. However, one crucial aspect of large language models has been neglected so far: the pretraining datasets. To address this gap, we present an open framework for the collection and systematic compilation of pretraining datasets, called LLM-Datasets. With LLM-Datasets, we make a community-effort and collaborate with experts from the individual languages to collect and systematically compile datasets suitable in terms of data quantity and quality for pretraining language models in a multilingual setting. The framework provides a unified interface to pretraining datasets enabling the download, text extraction, filtering, and sampling of the pretraining data. It is modular and extensible with new datasets and designed with high-performance-computing requirements in mind that are needed to achieve the scale of today's language models. Users of the framework can focus on the actual data composition and reuse existing datasets from the community while ensuring reproducibility. To showcase LLM-Datasets, we compiled a pretraining dataset with 2.3 trillion tokens for a large language model covering 32 European languages.

## 1 Introduction

Language models are the cornerstone of today's natural language processing research but their scale has made the training, evaluation, and deployment an increasingly complex project. The community reacted to the increasing complexity by developing dedicated tooling and framework for parts of the language model stack. For instance, DeepSpeed (Rasley et al., 2020), Megatron-LM (Shoeybi et al., 2019) or its forks (Black et al., 2022) are common codebases for large-scale model training. The lm-evaluation-harness (Gao et al., 2023) has become the defacto standard for benchmarking and evaluations. For inference and deployment, open source projects such as vLLM (Kwon et al., 2023) have been adopted by the community. Moreover, fine-tuning datasets are openly shared on the Hugging Face Hub (Lhoest et al., 2021). However, the pretraining datasets of language models and their processing pipelines are rarely shared with the community. Notable exceptions are Bloom (BigScience-Workshop et al., 2023), Pythia (Biderman et al., 2023), and OLMo (Groeneveld et al., 2024). Even language models that explicitly claim to be open, such as OPT (Zhang et al., 2022), do not provide sufficient details to reproduce the pretraining data. This gap is even more severe given that there is more and more evidence that the pretraining data is one of the primary factors that determine model quality (Gunasekar et al., 2023; Gemini-Team et al., 2023).

In this paper, we address the aforementioned gap and present LLM-Datasets as an open framework for pretraining datasets of language models. The remainder of this paper

describes the principles that guided the development of the framework, explains its components such as the processing pipeline and the dataset viewer, and finally showcases a pretraining dataset that was created with the framework. LLM-Datasets is publicly available as a Python package and its source code is on GitHub with an Apache-2.0 license.[1]

## 2  Related Work

Since the introduction of language models in the form of pre-trained fixed-word embeddings like word2vec (Mikolov et al., 2013), GloVe (Pennington et al., 2014) and fastText (Mikolov et al., 2018). Preprocessing the pre-trained data has become a crucial step in developing state-of-the-art language models, and while data demands have dramatically increased in recent years (Hoffmann et al., 2022), some of these fixed-word embeddings already used pre-training datasets in the Billions of tokens with Pennington et al. (2014) using a 840 billion token dataset based on Common Crawl[2] to train GloVe. And while data demands have increased to a point where we are beginning to question if we will run out of data (Villalobos et al., 2022) our data preprocessing pipelines have become more and more sophisticated throughout the years, and have started to target more diverse data sources in recent years.

One of the first data pipelines for processing pre-training data to be publicly available was that of fastText (Grave et al., 2018), this was a rather simple pipeline with very little documentation and based on bash scripts[3]. It was later with the release of OSCAR with its first pipeline *goclassy* (Ortiz Suárez et al., 2019) and CCNet (Wenzek et al., 2020) that pre-processing data pipelines became more central in scientific NLP publications.

From these first pipelines, up to more recent ones like C4's Raffel et al. (2020), Ungoliant (Abadji et al., 2021), Redpajama (Computer, 2023), RefinedWeb (Penedo et al., 2023), MADLAD (Kudugunta et al., 2023) and Datatrove (Penedo et al., 2024); the main focus have been on quickly processing, cleaning and filtering Web data, more precisely Common Crawl data, although some very recent projects have also used the Internet Archive as a source (Aulamo et al., 2023). Most of the differences between these pipelines and libraries now revolve around the infrastructure optimization (Abadji et al., 2021; Penedo et al., 2024), the choice of boilerplate removal tool (Pomikálek, 2011; Endrédy & Novák, 2013; Barbaresi, 2021), the language identification method (Grave et al., 2018; Caswell et al., 2020; Kreutzer et al., 2022; NLLB Team et al., 2022; Burchell et al., 2023), and the quality metrics or filters used to select which Web documents to retain (Heafield et al., 2013; Rae et al., 2021; Jansen et al., 2022; Caswell et al., 2023).

Since most modern models are now being trained with more than just Web data, recent initiatives have published data pipelines and management tools that can handle a wide variety of data sources. some of these include widely used data libraries like HF Datasets (Lhoest et al., 2021) but also data preparing and pre-processsing pipelines specifically targeted to the training of LLMs such as The Pile (Gao et al., 2020), ROOTS (Laurençon et al., 2022), DOLMA (Soldaini et al., 2024), the NeMo Data Curator[4]. Some data exploration tools have also been released at times along these pipelines (Piktus et al., 2023), and some newer initiatives like the Common Corpus[5] have moved away from using Web data altogether.

However, the aforementioned related work focuses either on techniques for data preprocessing or the collection of datasets with a narrow predefined scope, e.g., only English datasets. With LLM-Datasets, we combine both aspects and widen the scope of the data collection such that framework users can concentrate on the data composition and easily pick the most suitable datasets for their own LLM use case.

---

[1] https://github.com/malteos/llm-datasets
[2] https://commoncrawl.org
[3] https://github.com/facebookresearch/fastText/tree/main/crawl
[4] https://github.com/NVIDIA/NeMo-Curator
[5] https://huggingface.co/blog/Pclanglais/common-corpus

# 3 The Framework

## 3.1 Principles

The training of large language models is a complex project. To account to this complexity, the LLM-Datasets framework follows a set of key principles.

**HPC-ready.** Language models require large computing resources that are typically only available in dedicated high performance computing (HPC) facilities. We developed LLM-Datasets with the requirements of an HPC system in mind. For instance, data will most likely be stored in a network file system affecting the reading and writing of files. To account for this, all data produced by LLM-Datasets is stored in medium-sized chunks by default (10GB of uncompressed data). The file chunks are small enough to fit into memory (e. g., for shuffling) but at the same time they are large enough to reduce inode count.

**Modular and extensible.** The framework comes with a preconfigured collection of 2241 datasets. In addition to these datasets, users can extend the framework with their own custom datasets without modifying the main code base. We also provide abstractions for common datasets, e. g., Hugging Face datasets, that allow adding new datasets with a few lines of code. For all other data formats, only the text extraction logic needs to be implemented.

**Private data support.** While we are strong supporters of open data, we acknowledge that not all data can be shared openly due to license restrictions or other reasons. For this reason, the framework supports the use of private datasets. All user datasets can be assigned to specific registries and tagged according to their availability (public, on-request, or private).

**Reproducible dataset composition.** A key requirement for rigorous science is the reproduciblity of experiments. Thus, the framework is designed to produce the identical output dataset as long as the input configuration is identical. At the same time the underlying data may change during the development process, e. g., because novel data has become available. In LLM-Datasets, the final dataset composition is based on configuration files that enable the simple selection and sampling of underlying sources while being reproducible.

**Bring your own model.** Model architectures and implementations are evolving rapidly. To reflect this, LLM-Datasets is model agonistic, i. e., the composed and tokenized training data can be consumed via Python API or common modeling frameworks like Hugging Face Transformers (Wolf et al., 2020) or NVIDIA's Megatron-LM (Shoeybi et al., 2019).

## 3.2 Schema

Fig. 1 illustrates the schema in which information is organized in the framework. The central element is the *Dataset* class. It represents a specific implementation of one dataset. For example, all Wikipedia articles in a certain language from a specific date would correspond to one dataset. A *Dataset* is associated with attributes of metadata such as title, homepage, citation, or genre. However, more important are the attributes that are relevant for language modeling, i. e., the languages of the contained texts, license information, and the size of the dataset in terms of tokens and bytes. We also tag a *Dataset* for potential quality issues (e. g., short texts or encoding errors) and regarding their availability (public domain, on request etc.). All the metadata allows users selecting the datasets that are most suitable for their use case. Groups of datasets that share many attributes are combined into a *Data Source*, e. g., all Wikipedia versions are represented as a single data source. Besides the metadata, each *Dataset* implements a method that extracts the text from the raw dataset files. Taking again the example of Wikipedia: Extracting the plain text of Wikipedia's XML dumps would corresponds to this method. For common data formats or sources, text extraction methods are implemented and can be reused.

To make a *Dataset* accessible within the framework, it needs to registered in the *Dataset Registry*. The registry holds essential the information of all available datasets. If needed,

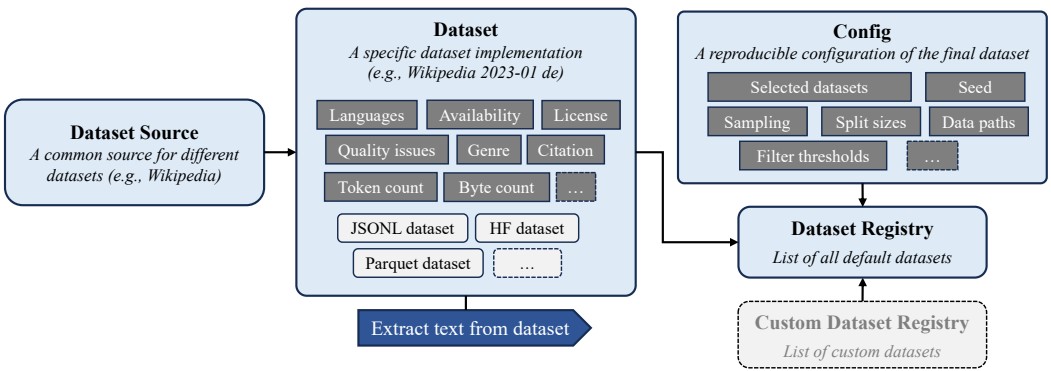

Figure 1: Each dataset is represented with a dataset class, which implements its own text extraction logic or inherits from common data sources or other abstractions, e. g., Hugging Face datasets. The final dataset composition is defined through reproducible configuration files with the information about the selection and sampling of individual datasets.

custom registeries can be defined, e. g., to have a clear distinction between public and private datasets. In addition to the data-centric elements, the *Config* is pivotal to the framework. A *Config* holds all information to make the creation of the final dataset reproducible and shareable such as random seeds, selected datasets, and sampling setting.

## 3.3 Processing Pipeline

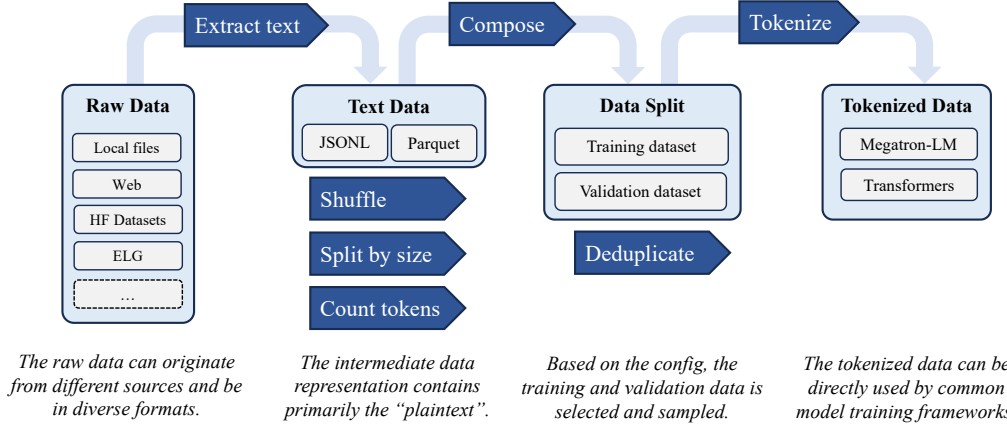

Figure 2: The framework's processing pipeline. The three main processing steps include the text extraction, data composition, and tokenization.

To obtain language data, which can be used for model training, the framework employs a pipeline with three main processing steps: text extraction, data composition, and tokenization. Fig. 2 summarizes the processing pipeline. Initially all data is considered as raw data, meaning the data resides in its original format as provided by the data source. The format in that raw data is stored can range from simple text files on the local file system or on the Web to PDFs or other formats that require sophisticated processing for the text extraction. As described in §3.2, each dataset class can implement its own independent text extraction step or reuse existing abstractions for common formats.

The extracted text is then stored in a format unified for all datasets, e.g., JSONL or Parquet. To reduce the file size, compression is enabled by default. At this stage, additional processing steps such as shuffling, counting tokens or splitting large files into small chunks can be performed. These processing steps are implemented but optional as they are not mandatory for all LLM use cases.

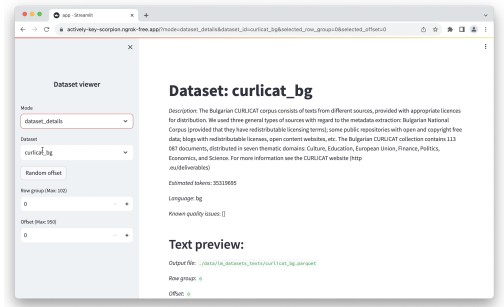

(a) Details, e.g., extracted texts.

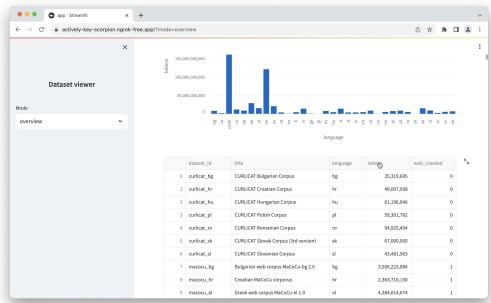

(b) Statistics, e.g, token count.

Figure 3: The framework comes with a Web-based dataset viewer that allows inspecting the text content of all datasets and provides an overview about statistics such as token count.

The next pipeline step is the data composition. The composition involves the mixing and sampling of individual datasets into one final dataset that can be split training and validation subsets. Configuration files (§3.2) define how the final dataset is composed. The composition step can be performed on-the-fly, i.e., the resulting samples are diretly consumed by a tokenizer, or stored on the file system. On top of the composed dataset, the framework can perform exact deduplication using locality sensitive hashing. As the final pipeline step, the composed data is fed into a tokenizer to produce the input data for the language model training framework. As tokenization frameworks, *Google SentencePiece*[6] and *Hugging Face tokenizers*[7] are implemented.

The framework intentionally does not implement common preprocessing techniques like near-deduplication or toxicity filters since other dedicated tools exist for these preprocessing steps, e.g., *text-dedup*[8] or *Perspective*[9]. However, these dedicated tools can be applied on the data files, either on the extracted texts or on the final composed dataset. For the *DataTrove* processing framework, we provide even a dedicated interface that allows ingesting all datasets implemented in LLM-Datasets with *DataTrove*.

A short documentation about installation, usage, and configuration of the framework can be found in Appendix A.1. For a more comprehensive overview and detailed examples, we refer to our Web-based documentation.[10]

## 3.4   Dataset Viewer

Training datasets of large language models are becoming so large that it is no longer feasible to inspect all the data manually. However, when deciding for data composition it is still crucial to manually look at least at a sample of the data. For this purpose, the framework comes with a Web-based data viewer that allows users to view the extracted text and statistics over the available datasets. With the dataset viewer, one can spot systematic errors that might occur during the processing of the datasets, e. g., bad encodings or false language identification. The dataset viewer is implemented as Streamlit application.[11]

---

[6] https://github.com/google/sentencepiece
[7] https://github.com/huggingface/tokenizers
[8] https://chenghaomou.github.io/text-dedup/
[9] https://perspectiveapi.com/
[10] https://malteos.github.io/llm-datasets/
[11] https://streamlit.io

# 4 A European Pretraining Dataset

The framework is language-agnostic and can be used for any kind of monolingual or multilingual pretraining datasets. As a first example, we compiled a pretraining dataset with 2.3 trillion tokens for a multilingual language model covering 32 European languages. The number of tokens is based on the Bloom tokenizer (BigScience-Workshop et al., 2023). We estimate the number of tokens on a dataset-level separately for each data subset by counting the white spaces in all documents, tokenizing 1,000 randomly selected documents, and then multiplying the white space count of all documents with the tokens per white space in the 1,000 documents. An overview of the European dataset is presented in the following.

## 4.1 Languages

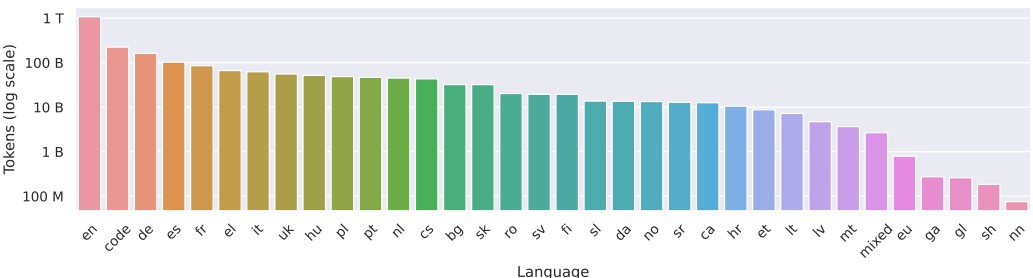

Figure 4: Tokens per language. English is by far the most dominant language.

The dataset contains text from the 24 official EU languages and eight additional regional and non-EU languages. Specifically, it covers English, German, Spanish, French, Greek, Italian, Ukrainian, Hungarian, Portuguese, Polish, Dutch, Czech, Slovak, Bulgarian, Swedish, Finnish, Romanian, Danish, Slovene, Norwegian, Croatian, Serbian, Estonian, Lithuanian, Catalan, Latvian, Maltese, Basque, Irish, Galician, Serbo-Croatian, Norwegian Nynorsk (sorted by number of tokens). Moreover, the dataset has a diverse set of programming languages, primarily from the StarCoder dataset (Li et al., 2023). Fig. 4 shows the distribution of tokens per language. English is by far the most dominant language, reflecting studies on the technology support of Europe's languages. There are also translation datasets that mix two languages (see "mixed" in Fig. 4).

A table with the token distribution by language can be found in the Appendix A.2. Please note that for a final data composition specific languages can be up- or down-sampled to achieve a more balanced language distribution tailored to the specific LLM use case.

## 4.2 Data Sources

The European pretraining dataset comprises 62 different data sources. A detailed list of all data sources can be found in Appendix A.3. The large number of different data sources is due to the diverse set of languages covered by the dataset.

Most of the data sources are community contributes, i.e., they were suggested by researchers from the individual language communities. When compiling the dataset, the initial goal was to have at least one dedicated monolingual data source per language and to not only rely on multilingual data sources such as Web-crawled data. The intuition behind this approach was that curated language-specific data will presumably have a higher quality compared to their multilingual counterparts. However, no such language-specific data sources could be integrated for low-resource languages like Irish. For these languages, the data originates only from multilingual sources.

The largest data source is Web-crawled data from Colossal OSCAR, separately described in §4.3. Other notable sources are the multilingual corpora from MaCoCu (Bañón et al.,

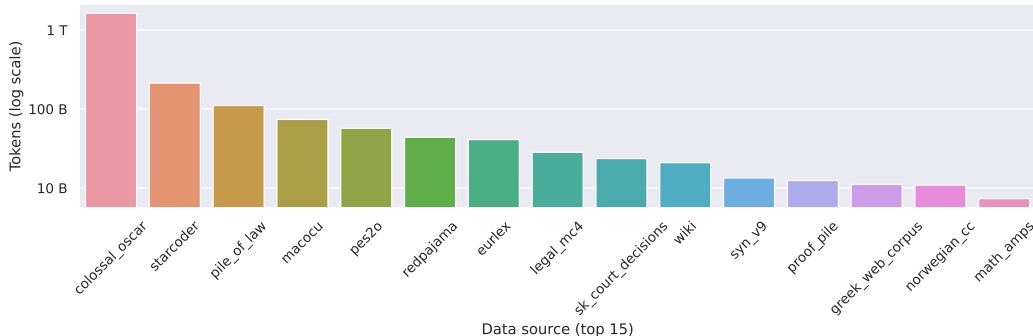

Figure 5: Tokens per data source for the top-5 sources. The largest data source is Web-crawled data from Colossal OSCAR.

2022), EURLex[12], and Wikimedia projects like Wikipedia, Wikinews, and Wikibooks. We also make use of high quality English sources. For instance, scientific literature from peS2o (Soldaini & Lo, 2023) or mathematical corpora like Proof Pile and AMPS Hendrycks et al. (2021). To avoid duplicated content, we discard specific subsets from data sources, e. g., only the StackExchange subset from RedPajama[13] is used and other RedPajama subsets such as Wikipedia or CommonCrawl are discarded. Other subsets like RedPajama-Books are removed due to legal reasons. In general, the European pretraining dataset has a higher share of legal literature (laws and court decisions) compared to related work since in many European countries this data is public domain and available in large quantities. For the curated data sources (non-Web-crawled content), no additional filtering was performed apart from removing texts with less than 450 characters.

## 4.3 Web-crawled Data from Colossal OSCAR

| Dump | Original docs. # [M] | Removed docs. by filter [%] | | | Filtered docs. | |
|------|------|------|------|------|------|------|
| | | Quality W. | Harmful$_{PP}$ | Categories | # [M] | Tokens [B] |
| 2015-14 | 388.9 | 89.91% | 0.67% | 1.94% | 38.2 | 102.9 |
| 2016-40 | 553.9 | 88.52% | 1.08% | 7.12% | 58.4 | 158.3 |
| 2017-43 | 1101.9 | 87.03% | 1.62% | 1.54% | 138.4 | 282.3 |
| 2018-47 | 768.8 | 84.99% | 1.43% | 0.50% | 113.1 | 242.6 |
| 2019-22 | 815.3 | 86.46% | 1.52% | 0.32% | 108.3 | 226.7 |
| 2020-24 | 814.0 | 87.94% | 2.18% | 0.32% | 95.7 | 209.6 |
| 2020-45 | 801.8 | 88.00% | 2.36% | 0.36% | 93.6 | 196.4 |
| 2021-49 | 760.1 | 87.40% | 3.35% | 0.23% | 92.3 | 197.9 |
| 2022-27 | 997.7 | 87.70% | 2.80% | 0.25% | 119.0 | 245.8 |
| 2022-49 | 1096.8 | 87.61% | 2.89% | 0.23% | 131.7 | 279.2 |
| 2023-14 | 1002.2 | 87.35% | 2.75% | 0.22% | 123.0 | 245.5 |
| 2023-23 | 1029.5 | 87.16% | 2.85% | 0.19% | 128.2 | 257.9 |

Table 1: Number of original documents, percentage of documents removed by *quality warnings*, by *harmful KenLM model perplexity* and by *Blocklist Category*. The filters are applied in sequential order. The table also shows final number of documents and and estimated tokens after filtering. All of the numbers are given by Common Crawl *dump*.

---

[12]https://huggingface.co/datasets/joelniklaus/eurlex_resources
[13]https://huggingface.co/datasets/togethercomputer/RedPajama-Data-1T

By far the largest data source is *Colossal OSCAR* (Abadji et al., 2022; Jansen et al., 2022) which is an annotated version of twelve Common Crawl *snapshots* (also called *dumps*), i. e., Web-crawled data using the Ungoliant pipeline Abadji et al. (2021). The OSCAR project (Open Super-large Crawled Aggregated coRpus) is an open source project that aims to provide Web-based multilingual textual resources. The project provides language identification and annotations for some Common Crawl snapshots releasing them for more than 150 languages while trying to optimize the quality of these annotations (Kreutzer et al., 2022).

For our framework, we decided to include the latest release of OSCAR, namely Colossal OSCAR 1[14]. The release comprises ten Common Crawl snapshots and uses the latest quality metrics introduced into the Ungoliant pipeline by Abadji et al. (2022) and Jansen et al. (2022). We also used OSCAR 2023.01[15] that uses the exact same version of Ungoliant as Colossal OSCAR 1, and we also run Ungoliant ourselves on the CC-MAIN-2023-14 to create one more OSCAR like dataset. However, OSCAR provides only annotations to the Common Crawl data. The decision of how to filter the data is left to the user. Therefore, we describe in the following what filter settings we used to derive our final dataset. All settings are found by manually inspecting samples of the data and deciding for the settings that produce to highest quality of content.

First of all, we extract only the European languages mentioned in §4.1 from OSCAR. We rely on the annotations made by the Ungoliant pipeline in separate steps. First, we use the *quality warnings* introduced by Abadji et al. (2021) including:

1. tiny: The document has a low ($< 5$) number of lines.

2. short sentences: The document has a high number ($> 50\%$) of short lines ($< 400$ bytes)

3. header: The document has a high number of short lines at its head, suggesting the presence of low quality content.

4. footer: The document has a high number of short lines at its tail, suggesting the presence of low quality content.

5. noisy: The document has a high percentage of punctuation ($> 50\%$)

We remove every document containing any of these warnings. To the remaining documents, we then apply the KenLM-based (Heafield et al., 2013) *harmful perplexity* filter introduced by Jansen et al. (2022). We remove all documents with a perplexity below 25 and above 100,000. Finally, we used the URL-based category blocklist annotations based on the Toulouse UT1 Blocklist[16], removing all of the remaining documents that are annotated with the following categories:

- aggressive
- adult
- cryptojacking
- dangerous material
- phishing
- warez

- ddos
- hacking
- malware
- mixed adult
- sect

Tab. 1 and Tab. 2 show the number of documents by both date of the CommonCrawl dump and language, the percentage of documents removed at each filtering step, and the final document and token estimate count after filtering.

---

[14]https://huggingface.co/datasets/oscar-corpus/colossal-oscar-1.0
[15]https://huggingface.co/datasets/oscar-corpus/OSCAR-2301
[16]http://dsi.ut-capitole.fr/blacklists/index_en.php

| Lang. | Original docs. | Removed docs. by filter [%] | | | Filtered docs. | |
|---|---|---|---|---|---|---|
| | # [k] | Quality W. | Harmful$_{PP}$ | Categories | # [k] | Tokens [M] |
| *Top-5* | | | | | | |
| en | 6,137,246.1 | 86.84% | 2.42% | 0.76% | 782015.3 | 950,517.47 |
| de | 841,750.9 | 88.19% | 5.56% | 0.19% | 93685.4 | 134,711.25 |
| es | 654,498.5 | 87.43% | 1.13% | 0.17% | 81177.7 | 89,646.34 |
| fr | 629,855.4 | 88.62% | 1.45% | 0.35% | 70387.0 | 78,929.48 |
| it | 340,549.9 | 89.96% | 0.33% | 0.19% | 34024.0 | 50,417.61 |
| *Bottom-5* | | | | | | |
| ga | 140.2 | 87.80% | 0.00% | 0.10% | 17.1 | 24.44 |
| nn | 73.9 | 95.27% | 0.00% | 0.00% | 3.5 | 2.00 |
| hr | 138.9 | 98.72% | 0.00% | 0.00% | 1.8 | 0.99 |
| mt | 21.7 | 98.02% | 0.00% | 0.00% | 0.4 | 0.41 |
| sh | 20.3 | 99.81% | 0.00% | 0.00% | 0.0 | 0.02 |

Table 2: Number of original documents, percentage of documents removed by *quality warnings*, by *harmful KenLM model perplexity* and by *Blocklist Category*. The filters are applied in sequential order. The table also shows final number of documents and estimated tokens. The statistics are computed for the top 5 and bottom 5 languages.

# 5    Conclusions

We have presented LLM-Datasets, an open framework for pretraining datasets of large language models. LLM-Datasets provides a unified interface for downloading, text extraction, filtering, and sampling of large text datasets, including additional tools such as a dataset viewer. At the time of writing, 2241 datasets including subsets from 62 sources in 164 languages are implemented in the framework and readily available for language model training. In this paper, we further showcase the framework by presenting a European pretraining dataset that represents a subset of all implemented datasets. The European dataset contains 2.3 trillion tokens of multilingual text, covers 32 European languages, and is fully reproducible with the LLM-Datasets framework. Parts of this dataset are already being used to train language models.[17]

### Acknowledgments

The work presented in this paper has received funding from the German Federal Ministry for Economic Affairs and Climate Action (BMWK) through the project OpenGPT-X (project no.68GX21007D). The majority of the work presented in this paper was carried out while all four authors were with DFKI, working on the OpenGPT-X project. Some of the work carried out by Pedro Ortiz Suarez was also done within Occiglot where he remains a member and active contributor.

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

# A  Appendix

## A.1  Usage example

**Getting started.**  The framework can be installed via the Python Package Index (PyPI):

```
pip install llm-datasets
```

After installing the Python package, the individual processing steps can be run via the command line.  For example, the download and extraction of plain-text of one or more datasets can be run with the following command:

```
llm-datasets extract_text $DATASET_ID $OUTPUT_DIR  [--output_format parquet] ...
```

**Configuration.**  LLM-Datasets allows you to specific general settings through config files so you do not need to specific always the same command line arguments. Several commands support passing the *–configs* argument which should point to one or more YAML-files on your file system. For example, the text extraction command:

```
llm-datasets extract_text ... --configs $PATH_TO_YAML_CONFIG_FILE
```

In the config files, you can store for example system specific settings like the local paths, where the raw dataset files are located:

```yaml
# ./examples/llm_datasets_configs/my_system.yaml
local_dirs_by_source_id:
  redpajama: /my_system_specific_data_directory/redpajama
```

**Dataset selection and sampling.**  The configuration files are also needed for specifying the final dataset composition, including the selection of the datasets and their sampling. The following examples shows a config for an Italian dataset:

```yaml
# ./examples/llm_datasets_configs/italian_data.yaml

# a fixed random seed for shuffling etc.
seed: 0

selected_dataset_ids:
  # italian subsets
  - itwac
  - eurlex_it
  - wikipedia_20231101_it
  - wikibooks_it
  - wikinews_it
  - colossal_oscar_2023-23_it
  - parlamint_it

# down-sample webcrawled + up-sampled high quality
sampling_factor_by_source_id:
  colossal_oscar: 0.1

sampling_factor_by_dataset_id:
  itwac: 0.5
  eurlex_it: 2
  wikipedia_20231101_it: 3
```

| Language | Estimated Tokens (M) | Percentage (%) |
|---|---|---|
| English (en) | 1,079,192 | 46.90 |
| Code (code) | 223,834 | 9.73 |
| German (de) | 161,521 | 7.02 |
| Spanish (es) | 102,586 | 4.46 |
| French (fr) | 85,394 | 3.71 |
| Greek (el) | 66,371 | 2.88 |
| Italian (it) | 62,408 | 2.71 |
| Ukrainian (uk) | 55,650 | 2.42 |
| Hungarian (hu) | 51,686 | 2.25 |
| Polish (pl) | 48,640 | 2.11 |
| Portuguese (pt) | 46,726 | 2.03 |
| Dutch (nl) | 45,090 | 1.96 |
| Czech (cs) | 43,286 | 1.88 |
| Bulgarian (bg) | 32,168 | 1.40 |
| Slovak (sk) | 32,145 | 1.40 |
| Romanian (ro) | 20,267 | 0.88 |
| Swedish (sv) | 19,458 | 0.85 |
| Finnish (fi) | 19,393 | 0.84 |
| Slovene (sl) | 13,726 | 0.60 |
| Danish (da) | 13,604 | 0.59 |
| Norwegian (no) | 13,387 | 0.58 |
| Serbian (sr) | 12,858 | 0.56 |
| Catalan (ca) | 12,582 | 0.55 |
| Croatian (hr) | 10,559 | 0.46 |
| Estonian (et) | 8,759 | 0.38 |
| Lithuanian (lt) | 7,239 | 0.31 |
| Latvian (lv) | 4,719 | 0.21 |
| Maltese (mt) | 3,668 | 0.16 |
| Mixed (mixed) | 2,665 | 0.12 |
| Basque (eu) | 792 | 0.03 |
| Irish (ga) | 274 | 0.01 |
| Galician (gl) | 260 | 0.01 |
| Serbo-Croatian (sh) | 184 | 0.01 |
| Norwegian Nynorsk (nn) | 76 | 0.00 |
| *Total tokens:* | 2,301,165 | 100.0 |

Table 3: Language distribution as millions of tokens (based on the Bloom tokenizer).

## A.2 Language distribution in the European dataset

The European dataset contains 1,138 datasets (including subsets) from 67 sources in 32 languages. The languages are as follows: Bulgarian, Catalan, Czech, Danish, German, Greek, English, Spanish, Estonian, Basque, Finnish, French, Irish, Galician, Croatian, Hungarian, Italian, Lithuanian, Latvian, Maltese, Dutch, Norwegian Nynorsk, Norwegian, Polish, Portuguese, Romanian, Serbo-Croatian, Slovak, Slovene, Serbian, Swedish, Ukrainian. Tab. 3 lists all languages and estimated number of tokens.

## A.3 List of data sources in the European dataset

LLM-Datasets implements different data sources ranging from Web-crawled data such as Colossal OSCAR to small curated datasets with high quality content. The data sources used in the European dataset are listed in Tab. 4 including estimated number of tokens.

Table 4: Available European data sources and their estimated token count.

| Title | Description | Tokens |
|---|---|---|
| CURLICAT Corpus (Váradi et al., 2022) | The CURLICAT corpus includes 7 monolingual corpora (Bulgarian, Croatian, Hungarian, Polish, Romanian, Slovak and Slovenian) containing selected samples from respective national corpora. | 963 M |
| MaCoCu web corpus (Bañón et al., 2022) | MaCoCu focuses on collecting monolingual and parallel data from the Internet, specially for under-resourced languages and DSI-specific data. See https://macocu.eu/ | 74 B |
| RedPajama-Data T1 (selected subsets) (Computer, 2023) | An Open Source Recipe to Reproduce LLaMA training dataset | 18 B |
| peS2o (Soldaini & Lo, 2023) | The peS2o dataset is a collection of ~40M creative open-access academic papers, cleaned, filtered, and formatted for pre-training of language models. It is derived from the Semantic Scholar Open Research Corpus(Lo et al, 2020), or S2ORC. | 57 B |
| Auxiliary Mathematics Problems and Solutions (AMPS) dataset (Hendrycks et al., 2021) | Our pretraining dataset, the Auxiliary Mathematics Problems and Solutions (AMPS) dataset, has problems and step-by-step solutions typeset in LATEX. AMPS contains over 100,000 problems pulled from Khan Academy and approximately 5 million problems generated from manually designed Mathematica scripts. Problems include various aspects of algebra, calculus, counting and statistics, geometry, linear algebra, and number theory. | 7 B |
| Open Legal Data - German court decisions and laws (Ostendorff et al., 2020) | OPENLEGALDATA.IO is a free and open platform that makes legal documents and information accessible to the public. | 7 B |
| DeWaC | DeWaC is a 1.7 billion word corpus constructed from the Web limiting the crawl to the .de domain and using medium-frequency words from the SudDeutsche Zeitung corpus and basic German vocabulary lists as seeds. | 3 B |
| The Gaois bilingual corpus of English-Irish legislation (Irish legislation) | Bilingual corpus of English-Irish legislation provided by the Department of Justice. | 4 k |
| Irish Universal Dependencies | Universal Dependencies (UD) is a framework for consistent annotation of grammar (parts of speech, morphological features, and syntactic dependencies) across different human languages. | 40 k |
| Croatian web corpus hrWaC 2.1 (Ljubešić & Klubička, 2014) | hrWaC is a web corpus collected from the .hr top-level domain. The current version of the corpus (v2.0) contains 1.9 billion tokens and is annotated with the lemma, morphosyntax and dependency syntax layers. | 2 B |
| ITWaC | itWaC: a 2 billion word corpus constructed from the Web limiting the crawl to the .it domain and using medium-frequency words from the Repubblica corpus and basic Italian vocabulary lists as seeds. | 3 B |

**Table 4 – continued from previous page**

| Title | Description | Tokens |
|---|---|---|
| Korpus Malti (Micallef et al., 2022) | General Corpora for the Maltese Language. This dataset is composed of texts from various genres/domains written in Maltese. | 816 M |
| SoNaR Corpus NC 1.2 | The SoNaR Corpus contains more than 500 million words from texts in standard Dutch later than 1954. All texts were tokenized, tagged for part of speech and lemmatized. The named entities were also labelled. All annotations were produced automatically, no manual verification took place. | 746 M |
| Corpus of academic Slovene KAS 2.0 (Žagar et al., 2022) | The KAS corpus of Slovene academic writing consists of almost 65,000 BSc/BA, 16,000 MSc/MA and 1,600 PhD theses (82 thousand texts, 5 million pages or 1,5 billion tokens) written 2000 - 2018 and gathered from the digital libraries of Slovene higher education institutions via the Slovene Open Science portal (http://openscience.si/). | 3 B |
| slWaC web corpus | slWaC is a web corpus collected from the .si top-level domain in 2011 and 2014. The corpus is tokenized and annotated with the lemma and the morphosyntax layer. | 3 B |
| od-justice 2.0 | Slovak court decisions. The corpus is based on data made available by the Ministry of Justice of the Slovak Republic. | 24 B |
| Korpus slovenských právnych predpisov v1.9 | Slovak body of laws (1955-2022) | 105 M |
| SYN v9: large corpus of written Czech (Křen et al., 2021) | Corpus of contemporary written (printed) Czech sized 4.7 GW (i.e. 5.7 billion tokens). It covers mostly the 1990-2019 period and features rich metadata including detailed bibliographical information, text-type classification etc. SYN v9 contains a wide variety of text types (fiction, non-fiction, newspapers), but the newspapers prevail noticeably. | 13 B |
| Danish GigaWord (Strømberg-Derczynski et al., 2021) | A billion-word corpus of Danish text. Split into many sections, and covering many dimensions of variation (spoken/written, formal/informal, modern/old, rigsdansk/-dialect, and so on).The license is CC-BY 4.0, Creative Commons with Attribution. Owners: ITU; Leon Derczynski, Manuel R. Ciosici | 2 B |
| DaNewsroom (Varab & Schluter, 2020) | A Large-scale Danish Summarisation Dataset | 835 M |
| DK-CLARIN Reference Corpus of General Danish | Reference Corpus of General Danish | 80 M |
| CaBeRnet: a New French Balanced Reference Corpus (Popa-Fabre et al., 2020) | A new balanced French corpus, CaBeRnet, that features a representative range of language usage, including a balanced variety of genres (oral transcriptions, newspapers, popular magazines, technical reports, fiction, academic texts), in oral and written styles. | 599 M |

**Table 4 – continued from previous page**

| Title | Description | Tokens |
|---|---|---|
| Norwegian Colossal Corpus | The Norwegian Colossal Corpus is a collection of multiple smaller Norwegian corpuses suitable for training large language models. We have done extensive cleaning on the datasets, and have made them available in a common format. The total size of the NCC is currently 45GB. Documents: 20,830,348; Words/document: 331 | 11 B |
| NKJP-PodkorpusMilionowy-1.2 (National Corpus of Polish) | A reference corpus of Polish language containing over fifteen hundred millions of words. The list of sources for the corpora contains classic literature, daily newspapers, specialist periodicals and journals, transcripts of conversations, and a variety of short-lived and internet texts. | 3 M |
| Polish Parliamentary Corpus / Korpus Dyskursu Parlamentarnego | The Polish Parliamentary Corpus (PPC) is a Polish corpus made up of documents from the proceedings of the Polish Parliament, Sejm, and Senate. The corpus includes data of the Polish Sejm corpus and consists of stenographic records of plenary sittings and committee sittings, segments of interpellations and questions. Texts in the PPC corpus cover the period of a hundred years from 1919 to 2019. | 1 B |
| ParlamentoPT | The ParlamentoPT is a Portuguese language data set obtained by collecting publicly available documents containing transcriptions of debates in the Portuguese Parliament. The data was collected from the Portuguese Parliament portal in accordance with its open data policy. | 732 M |
| Brazilian Portuguese Web as Corpus (Wagner Filho et al., 2018) | The BrWaC (Brazilian Portuguese Web as Corpus) is a large corpus constructed followingthe Wacky framework, which was made public for research purposes.The current corpus version, released in January 2017, is composed by 3.53 million documents,2.68 billion tokens and 5.79 million types. | 4 B |
| Bilingual English-Lithuanian parallel corpus from Seimas of the Republic of Lithuania website | Contents of http://www.lrs.lt were crawled, aligned on document and sentence level and converted into a parallel corpus. | 12 k |
| Corpus of State-related content from the Latvian Web (Processed) | Latvian Web, home pages of ministries and state public services, army, etc. were crawled, and parallel Latvian-English content was collected. | 52 k |
| Greek Legal Code (Papaloukas et al., 2021) | Greek_Legal_Code (GLC) is a dataset consisting of approx. 47k legal resources from Greek legislation. The origin of GLC is "Permanent Greek Legislation Code - Raptarchis", a collection of Greek legislative documents classified into multi-level (from broader to more specialized) categories. | 80 M |
| Greek Web Corpus (Outsios et al., 2018) | A corpus of the Greek Web used for training 'GreekBART: The First Pretrained Greek Sequence-to-Sequence Model' | 11 B |

*Continued on next page*

**Table 4 – continued from previous page**

| Title | Description | Tokens |
|---|---|---|
| Estonian Reference Corpus | This corpus includes Estonian texts (fiction, PhD theses, newspapers, magazines, parliamentary transcriptions, computer-mediated communication) published between 1990 and 2007. The corpus is encoded in TEI. The corpus is available for online browsing through a dedicated concordancer and is available for download from CELR. | 481 M |
| Estonian National Corpus 2021 | Corpus is based on Estonian National Corpus 2013, which was renewed by Lexical Computing Ltd. in 2017 and 2019 at the request of Estonian Language Institute.Subcorpora are: Estonian Reference Corpus 1990-2008, Estonian Web 2013, Estonian Web 2017, Estonian Web 2019, Estonian Wikipedia 2017, Estonian Wikipedia 2019, Estonian Open Access Journals (DOAJ), blogs, discussion, education, fiction, food, health, journals, news, religion, science, sex, society, sports. | 3 B |
| EusCrawl (filtered: no Wikipedia, no NC-licenses) (Artetxe et al., 2022) | EusCrawl is a high-quality corpus for Basque comprising 12.5 million documents and 423 million tokens, totalling 2.1 GiB of uncompressed text. EusCrawl was built using ad-hoc scrapers to extract text from 33 Basque websites with high-quality content, resulting in cleaner text compared to general purpose approaches. | 335 M |
| Spanish Legal Domain Corpora (Gutiérrez-Fandiño et al., 2021) | A collection of corpora of Spanish legal domain. | 1 B |
| Yle Finnish News Archive | The corpus, containing the articles from YLE https://yle.fi from 2019 and 2020, is available at www.kielipankki.fi/download | 286 M |
| The Swedish Culturomics Gigaword Corpus | One billion Swedish words from 1950 and onwards. Code to extract data from the corpus, as well as usage instructions, can be downloaded from https://svn.spraakdata.gu.se/sb-arkiv/tools/gigaword/ | 528 M |
| SrpKorSubset (news, legal, academic, conversation, literary) | The Corpus of contemporary Serbian, SrpKor, consists of 4,925 texts. | 866 M |
| MARCELL Romanian legislative subcorpus v2 | The Romanian corpus contains 163,274 files, which represent the body of national legislation ranging from 1881 to 2021. This corpus includes mainly: governmental decisions, ministerial orders, decisions, decrees and laws. All the texts were obtained via crawling from the public Romanian legislative portal. | 1 B |
| Corpus of laws and legal acts of Ukraine | A large (more than 9 Gb) corpus of laws and legal acts of Ukraine. | 2 B |
| EurlexResources | A Corpus Covering the Largest EURLEX Resources. | 41 B |
| LegalMC4 | MC4_Legal: A Corpus Covering the Legal Part of MC4 for European Languages | 28 B |

**Table 4 – continued from previous page**

| Title | Description | Tokens |
|---|---|---|
| Wikipedia | The free encyclopedia that anyone can edit. | 21 B |
| Wikibooks | The open-content textbooks collection that anyone can edit. | 313 M |
| Wikiquote | The free quote compendium that anyone can edit. | 247 M |
| Wikinews | News written by volunteers. | 90 M |
| Wikisource | The free library that anyone can improve. | 2 B |
| Wikivoyage | The free worldwide travel guide that you can edit. | 119 M |
| Colossal OSCAR 1 (Jansen et al., 2022) | The OSCAR project (Open Super-large Crawled Aggregated coRpus) is an Open Source project aiming to provide web-based multilingual resources and datasets for Machine Learning (ML) and Artificial Intelligence (AI) applications. The project focuses specifically in providing large quantities of unannotated raw data that is commonly used in the pre-training of large deep learning models. | 2 T |
| Starcoder (Li et al., 2023) | The dataset used for training StarCoder and StarCoder-Base. It contains 783GB of code in 86 programming languages, and includes 54GB GitHub Issues + 13GB Jupyter notebooks in scripts and text-code pairs, and 32GB of GitHub commits, which is approximately 250 Billion tokens. | 212 B |
| Pile of Law (Henderson* et al., 2022) | We curate a large corpus of legal and administrative data. The utility of this data is twofold: (1) to aggregate legal and administrative data sources that demonstrate different norms and legal standards for data filtering; (2) to collect a dataset that can be used in the future for pretraining legal-domain language models, a key direction in access-to-justice initiatives. | 34 B |
| CATalog 1.0 (without OS-CAR subsets) | CATalog is a diverse, open-source Catalan corpus for language modelling. It consists of text documents from 26 different sources, including web crawling, news, forums, digital libraries and public institutions, totaling in 17.45 billion words. | 7 B |
| Open Discourse Bundestag (Richter et al., 2020) | Open Discourse is the first fully comprehensive corpus of the plenary proceedings of the federal German Parliament (Bundestag). | 442 M |
| Tagesschau Archive Article Dataset | A scrape of Tagesschau.de articles from 01.01.2018 to 26.04.2023. Find all source code in github.com/bjoernpl/tagesschau. | 29 M |
| PhilPapers (2023 version) | A filtered version of the open access collection of philosophy publications PhilPapers, data-ready for The-Pile. | 911 M |
| Project Gutenberg books published before 1919 (Rae et al., 2019) | | 3 B |

**Table 4 – continued from previous page**

| Title | Description | Tokens |
|---|---|---|
| BigPatent (Sharma et al., 2019) | BigPatent consisting of 1.3 million records of U.S. patent documents along with human written abstractive summaries. | 8 B |
| Proof-Pile-2 (algebraic-stack) (Azerbayev et al., 2023) | AlgebraicStack, an 11B-token dataset of source code from 17 languages, spanning numerical, symbolic, and formal math. The dataset consists of filtered code from the Stack (Kocetkov et al., 2022), public GitHub repositories, and formal proofstep data | 23 B |
| Swiss-Judgment-Prediction (Niklaus et al., 2021) | Swiss-Judgment-Prediction is a multilingual, diachronic dataset of 85K Swiss Federal Supreme Court (FSCS) cases | 55 M |
| Swiss Legislation (Rasiah et al., 2023) | Swiss Legislation is a multilingual, diachronic dataset of 36K Swiss laws. This dataset is part of a challenging Information Retreival task. | 227 M |
| Tatoeba (Tiedemann, 2012) | Tatoeba is a collection of sentences and translations. | 7 M |
| OPUS-100 (Zhang et al., 2020) | OPUS-100 is English-centric, meaning that all training pairs include English on either the source or target side. | 901 M |
| WMT19 (Workshop on Statistical Machine Translation) (Barrault et al., 2019) | Shared Task: Machine Translation of News. | 1 B |
| Opensubtitles (Tiedemann, 2016) | OpenSubtitles.org provides a large collection of user contributed subtitles in various languages for movies and TV programs. | 30 B |
| ParlaMint (Erjavec et al., 2023) | ParlaMint 4.0 is a set of comparable corpora containing transcriptions of parliamentary debates of 29 European countries and autonomous regions, mostly starting in 2015 and extending to mid-2022. | 2 B |

