# OpenReview forum: "LLM-Datasets: An Open Framework for Pretraining Datasets of Large Language Models"
_colmweb.org/COLM/2024/Conference — COLM_

### Official Review · Reviewer_WmR4 · 2024-05-09

**Rating:** 5
**Confidence:** 4
**Ethics Flag:** 1

**Summary:**

This paper makes the following contributions:
- LLM-Datasets, a open-source python package to facilitate the creation large-scale pre-training data for LLMs, with unified interfaces to handle the loading and processing of datasets of diverse sources
- The authors apply LLM-Datasets to create a large corpus of 2.3T trillion tokens covering 32 European Languages.

**Reasons To Accept:**

This paper is nicely written and the proposed library is simple and clear.

1. The unified framework in LLM-Datasets (illustrated in figure 1 and 2) can help unify datasets of many different sources to create a larger corpus for training LLMs. The clear characterization of the needed steps can streamline the data preprocessing and makes it easier to adopt.
2. The proposed dataset artifact can also be a helpful resource to the community, albeit missing some ablation studies/analysis that can help us better understand the dataset as well as how it helps the actual training of an LLM.

**Reasons To Reject:**

Albeit the simple and clear design of LLM-Datasets, it seems the library is missing some of the key functionalities for data engineering for LLM pretraining; also the paper misses details about the proposed python library.
1. **Understanding the overlapping and enabling deduplication among data sources**: Deduplicating training entries in datasets is crucial to the training data quality. In the case of LLM-Datasets, since its primary goal is to mix multiple sources of datasets, it would be helpful if the tool can help identify content overlapping among the sources, and support deduplication.  While in figure 2 and Sec 3  the author mentioned that LLM-Datasets supports the use of external tools to deduplicate data, it does not specifically address the need for cross-dataset duplication analysis and removal, which can be a great/desired component for such a library.
2. **Supporting ablation of different data mixtures**: another major recent finding is that proper mixture of datasets of different sources can have a major impact in LLM training. While currently LLM-Datasets can help mix datasets of different sources, there is no direct support to help understand which subsets of mixture can lead to desired model performance, which should be an important function for such a library.
3. **Lack of details about the implementation and computational efficiency**: processing trillion token data involves a lot of compute, memory, and storage. The author barely mentions the underlying implementation of the proposed library (only vague descriptions like “LLM-Datasets is publicly available as a Python package” (page 2) or “We developed LLM- Datasets with the requirements of an HPC system in mind” (page 3)); neither does the author specify the required computation cost (# of CPUs, peak memory, etc) for the created dataset of 2.3T tokens. Having a fast and efficient implementation should be a desired (and perhaps necessary) property of such a library; adding additional discussion (and, ideally, comparisons to libraries like [datatrove](https://github.com/huggingface/datatrove)) can strengthen the paper significantly.
4. **Examples of the APIs (or actual code) for LLM-Datasets usage**: while the architecture design of the proposed library is clearly documented (in fig 1 and 2), this work does not mention specific use / APIs of LLM-Datasets in detail. Having the python library as the primary contribution of this paper, it’s important to detail not only design but also concrete implementations of the tool in the paper (at least in the appendix).
5. **Missing in-depth analysis / ablation studies for the proposed dataset**: this paper details the processing pipeline for the processed dataset, however it is unclear whether the mixture is meaningfully helpful for LLM training – as mentioned above, fine-grained analysis in terms of the data mixtures ratios/duplication among datasets is essential, and ideally one can show the benefits of the dataset via training smaller/proxy LLMs that can help extrapolate the performance at a larger scale.

---

> ### Author Rebuttal · Authors · 2024-05-31
>
> Thank you for the review and the thoughtful comments about our paper.
>
> @Reasons To Reject:
>
> R1: The current release of the framework already supports exact deduplication and the next release will integrate all processing steps from Huggingface's recently published DataTrove framework, which also includes near deduplication.
>
> R2: As mentioned in R3@Reviewer vKjo, we consider this out of scope for this paper due to the computational cost associated with training a multilingual LLM, even smaller ones, but are working on this already for a future release of our framework, and we expect support from the open source community in order to perform said ablations in the future. Probably the interesting thing that our framework can offer right now in this regard is the ease of reproducibility and of easily making changes to the data mixture, making ablations far more easier to perform for researchers and users of our framework.
>
> R3: Our framework relies on big-data technologies like parquet, which allows us to make everything as efficient in terms of I/O as possible. We will add the details of the infrastructure we used and the processing times for our 2.3T tokens dataset if our paper is accepted.
>
> R4: Please take a look at the example config (https://pastebin.com/4q5L4nSX) or the anonymized repo linked in the paper, more details will be also added to the appendix if the paper is accepted.
>
> R5: As mentioned above we consider this out of scope for this paper due to the computational cost associated with training a multilingual LLM, even smaller ones, but are working on this already for a future release of our framework, and we expect support from the open source community in order to perform said ablations in the future.

---

> > ### Comment · Reviewer_WmR4 · 2024-06-06
> > **Thank you for the rebuttal!**
> >
> > Thank you very much for the rebuttal! Given the rebuttal, I intend to keep my scores unchanged as some of my concerns are not resolved. I agree this library can be a helpful artifact for the community, but I am not convinced of the research value given the current scoping of the software /tool.

---

> ### Comment · Area_Chair_DZzH · 2024-06-05
> **Please respond to rebuttal and other reviewers' comments**
>
> Reviewer WmR4, could you please reply and/or update your review given the authors' response and discussion with the other reviewers? Thanks!

---

### Official Review · Reviewer_vKjo · 2024-05-11

**Rating:** 6
**Confidence:** 3
**Ethics Flag:** 1

**Summary:**

This work seeks to address an important problem: the lack of public disclosure and documentation surrounding pretraining datasets. It is largely well-written, and the authors have considered several desiderata for such a framework. However, I found that the central motivation of the paper needs to be clarified more carefully (see my comments in questions to authors), and some key claims need to be justified better (such as those surrounding reproducibility of pretraining datasets, see my questions to authors).

EDIT: I am updating my score to 6 on the basis of the author response

**Questions To Authors:**

(1) Could the authors clarify how their framework can improve the disclosure of pretraining data, by entities who do not currently disclose the data used to train their LLMs?

(2) Could the authors please provide more details on what an input configuration might look like?

(3) The authors state that “the framework is designed to produce the identical output dataset as long as the input configuration is identical”. Does this mean that it is possible to reproduce the dataset with exactly the same documents (and not just the correct mixture of tokens from similar sources)? If so, how detailed does the input configuration have to be to support this? If not, could the authors comment on the exact reproducibility of the dataset?

(4) Are there plans to update the datasets in LLM-datasets? How frequently will these updates happen?

(5) Are the different data sources stratified by language or type? For example, are wikipedias of all different languages considered a single data source within the framework?

(6) Minor writing suggestion: in section 4.1, how is ‘code’ in “from the 24 official EU languages and eight additional regional and non-EU languages. Specifically, it covers English, Code” different from “Moreover, the dataset has a diverse set of programming languages, primarily from the StarCoder dataset”. It was also odd to see code in the first sentence in the context of human languages.

(7) Minor suggestion: It would be quite useful to include recipes for standard quality filters within the framework as well, such as for near-deduplication, toxicity and PII (though the authors have justified their decision to not do this in the paper).

**Reasons To Accept:**

(1) A centralized framework that makes it easy to construct pretraining corpora is likely to be useful to the research community and I am supportive of the goals of this paper.

(2) The paper itself is fairly clear and well-written.

(3) I appreciate that the authors included a link to an anonymous github repository, which was very helpful in understanding the framework.

(4) The authors use their framework to construct a sizable multilingual pretraining corpus.

**Reasons To Reject:**

(1) Some of the main claims of this work are not well-motivated in the manuscript. For example “However, the pretraining datasets of language models and their processing pipelines are rarely shared with the community…In this paper, we address the aforementioned gap”. It is unclear to me how the contributed framework could help facilitate the release of pretraining datasets by entities who do not release them currently.

(2) I am not entirely sure that the pretraining datasets will be truly reproducible (see my question to authors), but will be willing to update this assessment based on the author response,

(3) It would have been nice to get an idea of the quality of the pretraining datasets it is possible to construct using this framework. I appreciate the authors were able to construct a large multilingual pretraining dataset, but it would have also been nice to get an idea of the quality of possible pretraining datasets.

---

> ### Author Rebuttal · Authors · 2024-05-31
>
> @Reasons To Reject:
>
> R1: Our audience are researchers who want to conduct open LLM research - not the ones who want to remain closed - and help them make their data work more reproducible by having standardized preprocessing for a large number of datasets. Moreover, we have focused on multilingual content, specially on low-resource European languages. As such we have curated and developed text extraction pipelines for datasets that were previously published only through country-specific platforms, we thus believe that we have made this resource far more discoverable and accessible.
>
> R2: We provide cleaning and text extraction scripts that are reproducible. The norm right now is that most LLM projects tend to use the same data sources but use different processing pipelines, this makes it hard to directly compare two modes that have been trained on the same data source as the actual processed training data might be different. Our goal with reproducibility is that researchers can report what dataset they processed with LLM-Datasets (our framework) and other researchers will be able to get the exact same data.
>
> R3: All datasets were manually inspected for their quality with the provided “dataset viewer”, which enables this. However, we acknowledge that the manual inspections do not necessarily correlate with the “data quality in terms of model performance”. Model training and evaluation are required to truly judge the quality of the data. We consider this out of scope for this paper due to the computational cost associated with training a multilingual LLM, but are working on this already.
>
> @Questions:
>
> Q1: See R1.
>
> Q2: https://pastebin.com/4q5L4nSX (will be added to Appendix)
>
> Q3:  Yes, it is possible to reproduce the dataset with exactly the same documents. See config config above.
>
> Q4: Yes, it is an ongoing effort, we also review and accept pull requests.
>
> Q5: Monolingual data sources are stratified by language, but datasource like Wikipedia or OSCAR are viewed as multilingual. However, our framework also allows to extract specific languages from those multilingual sources, that is, effectively restricting them to one language if the user so desires it.
>
> Q6:  Thank you, we will make the changes for the camera-ready version of the paper, if it is accepted.
>
> Q7: We are currently working on this and we will include some of these more sophisticated quality filters in future versions of the framework. See also R1 for reviewer 1.

---

> ### Comment · Reviewer_vKjo · 2024-06-04
>
> Thank you for the response authors!
>
> 1. I was making a writing suggestion that in the introduction, language such as "However, the pretraining datasets of language models and their processing pipelines are rarely shared with the community…In this paper, we address the aforementioned gap” is clarified as this might lead a reader to believe that this work is helping shed light on training datasets which are not released. The way I understand the contribution of this work is that though people may even release their training data, it is difficult to standardize and compare these different pretraining datasets, and the proposed framework facilitates that.
>
> 2. MINOR CLARIFICATION: On reading the responses to reviewers, I think the contribution of this project will largely depend on including high-quality data sources, and updating the store with new data sources that the community finds important. Would it be possible for the authors to share this plans about this going forward?

---

> > ### Author Response · Authors · 2024-06-06
> >
> > Thank you for your comment,
> >
> > 1. We will rephrase this in so that it is completely clear that we do not intend to shed light on training datasets which are not released, as we do not posses this information.
> >
> > 2. We plan to mantain and update the update the framework regularly. Regarding new data sources, since the submission of the current paper we have been working with our community in order to find and add new sources to the framework, in the future we expect to release new versions with new high quality sources with a specific emphasis in underrepresented and under resourced languages. We also expect to release further improvements on the filtering and cleaning scripts for the existing sources in order to produce even higher quality end-datasets.

---

> > > ### Comment · Reviewer_vKjo · 2024-06-06
> > >
> > > Thank you authors for your response!
> > >
> > > 1. No need to add a disclaimer about this, I was only suggesting to improve the writing and make it more precise so that the main contributions of the paper are clear.
> > > 2. This is good to know, thank you for sharing this.
> > >
> > > On the basis of the author response, I have increased my score to 6.

---

### Official Review · Reviewer_pGJL · 2024-05-11

**Rating:** 6
**Confidence:** 3
**Ethics Flag:** 1

**Summary:**

This work develops a framework to compile and collect pretraining datasets for large language models. They call their framework LLM-Datasets. LLM-Datasets contains an interface that makes downloading, extracting text, filtering, and sampling of pretraining data possible. They also talk about how modular it is: aiding in quick adaptability and extensibility to new data sources. Finally they use the framework to curate a 2.3T token dataset covering 32 european languages to demonstrate efficacy of LLM-Datasets. The framework claims to be HPC-ready, modular, supportive of private data, reproducible, and model agnostic.

**Questions To Authors:**

1) If you were trying to create some ideal pretraining corpus for an LLM using your framework and other tools that exist out there, what additional value add does the current instantiation of your framework have? This is the question that I'm confused about that would really help me evaluate the work more clearly.

2) How easy would it be for the framework to integrate existing toolkits such that you can use the framework to easily create your dataset that uses a specific config that you like on an already existing dataset? For example, if I wanted to use corpus X and use filters A, B, C from some toolkit on corpus X. It'd be awesome if I could press a couple of buttons and the framework would allow me to create that.

3) Is my question 2 beyond the scope of your project, where you simply are looking at creating a reproducible hub basically of datasets that can easily just be used to train LMs?

**Reasons To Accept:**

1) The problem that the authors are trying to solve is motivated adequately. It's clear to me that as we train more and more LLMs, having a unified, easily accessible and extensible framework for pretraining data is crucial.

2) The framework addresses some of the key aspects that such a framework needs to have, such as modularity and extensibility and private data support.

3) The adoption or integration of already created datasets into the framework is sizable. 2241 datasets including subsets from 62 sources in 164 languages is quite a lot of data!

**Reasons To Reject:**

1) The curation and filtering of the dataset ignores some filters that some recent datasets contain such as deduplication and toxicity filters. The authors say that tools already exist for these, but it seems like those were not utilized in the creation of this dataset. Personal information filters and other data contamination filters and things would be good to integrate such as those from Dolma (Soldaini et al. 2024).

2) Its not clear to me how much additional value this framework creates on top of existing toolkits if most of the filters are not native to the framework.

---

> ### Author Rebuttal · Authors · 2024-05-31
>
> Thank you for the review and the thoughtful comments about our paper.
>
> @Reasons To Reject:
>
> R1:  Our framework is designed with modularity in mind, as such, additional preprocessing steps from other open source projects can be easily integrated. For instance, the next release will include an `LLMDatasetsReader` for DataTrove (a data processing framework recently released by Huggingface) that would allow users to apply all filters from DataTrove on the datasets integrated into our framework.
>
> R2: The value lies in the integration of many different data sources that makes all the datasets available via a unified interface. Users do not need to implement the text extraction logic and cleaning themselves, which is a time consuming but necessary work. Users can instead focus on finding the best dataset composition for their own LLM use case. For our framework, we focused on providing text extraction and filtering for some datasets in underrepresented languages, as such, our framework provides tools to clean and extract texts from some sources that were not previously available on any other framework.
>
> @Questions:
>
> Q1: The value is the reduced effort in data sourcing (select the datasets that are relevant your use LLM use case), integrated datasets come with relevant information (license, language, size, what other models used this data already …). Also, as mentioned above, we focused on multilinguality and we cover dataset and sources that were not previously easily usable by researchers wanting to train LLMs in underrepresented languages.
>
> Q2: The modularity of our framework allows the datasets being processed by other tools, for example, with DataTrove (see R1). Moreover, the intermediate datasets (after text extraction) are stored in common file formats (Parquet or JSONL) that many other tools support, making cross-compatibility easy.
>
> Q3: A hub would be out of scope for our project however on the long run every useful (apart from existing hubs like the one from Huggingface). We primarily want to avoid repeated work and make it easier for researcher to compile better datasets for LLM pretrainingsince we see this as the major weakness of current academic LLM projects. Also as mentioned previously, we also focused on multilinguality and worked on some underrepresented European languages, as we want to provide easy to use datasets for practitioners wanting to develop these technologies in their own languages.

---

> > ### Comment · Reviewer_pGJL · 2024-06-05
> > **Response to rebuttal**
> >
> > Thanks for the rebuttal. You've addressed the questions and concerns as best as you could given the scope of the paper. I was between a 5 and a 6 last time and I think I'm solidly at a 6 after reading the rebuttal. It's hard for me to gauge the utility of the work, especially because it feels more like an ongoing effort that's going to improve over time and integrate more sources. My score remains unchanged at a 6.

---

### Decision · Program_Chairs · 2024-07-10

**Decision:**

Accept

**Comment:**

In this paper the authors propose an open source software framework, LLM-Datasets for collecting pretraining data for LLMs. Their pipeline consists of downloading raw web documents, extracting text, filtering, and subsampling examples, and the authors also include a Streamlit frontend interface for data visualization. They demonstrate the framework by using it to collect a 2.3T token dataset that includes data from 32 European languages.

Reviewers agreed that the paper was well written, the new dataset represents a useful contribution (though perhaps could use more analysis/study), and that having a unified framework for pertaining data curation could help alleviate some of the challenges in pretraining data curation, notably duplicated engineering effort and reproducibility. Of course, in order for the tool to serve these purposes, it must be highly flexible and easy to use. Reviewers expressed concerns about the maturity of the tool in terms of key functionality such as easily incorporating existing and new filters and data sources, and efficiency optimizations. The authors noted that they are working on adding functionality that would allow for running any filters available through DataTrove, but it remains unclear how easily the framework can be updated to include similar functionality on an ongoing basis.